# The Use of Targeted Monoclonal Antibodies in the Treatment of ABPA—A Case Series

**DOI:** 10.3390/medicina58010053

**Published:** 2021-12-29

**Authors:** Aoife O’Reilly, Eleanor Dunican

**Affiliations:** 1Department of Respiratory Medicine, St. Vincent’s University Hospital, Elm Park, D04 T6F4 Dublin, Ireland; eleanor.dunican@ucd.ie; 2School of Medicine, University College Dublin, Belfield, D04 V1W8 Dublin, Ireland

**Keywords:** allergic bronchopulmonary aspergillosis (ABPA), asthma, *Aspergillus fumigatus*, monoclonal antibodies, immunoglobulin E

## Abstract

Allergic bronchopulmonary aspergillosis (ABPA) is a pulmonary disorder occurring in response to *Aspergillus fumigatus* that can complicate the course of asthma and cystic fibrosis. Here we present a case of acute ABPA without central bronchiectasis, a case of chronic active ABPA with central bronchiectasis, and a case of severe relapsing ABPA with central bronchiectasis. All three were initially treated with corticosteroids and antifungal agents but had an incomplete response. These patients were then treated with anti-IgE therapy with omalizumab before being switched to the anti-IL5R agent benralizumab. They responded well to both agents. These case reports highlight the potential role of omalizumab and benralizumab in the treatment of ABPA, but further studies are required to evaluate the effectiveness of these medications. Longer follow-up periods and objective measurements of the impact of treatment are necessary.

## 1. Introduction

Allergic bronchopulmonary aspergillosis (ABPA) is an immunologically mediated pulmonary disorder that can complicate the course of asthma and cystic fibrosis. It is a hypersensitivity reaction mediated by antigens released by *Aspergillus* (most commonly *Aspergillus fumigatus*), which colonize the airways of susceptible subjects. ABPA is underdiagnosed, and there is a lack of widespread community-based data on its prevalence [1]; however, it has been estimated that between 1% and 5% of asthmatics develop ABPA [1]. The prevalence is reported to be higher in patients attending severe asthma clinics, at 13% [2]. A high level of suspicion for ABPA is crucial, particularly in those who present with frequent acute exacerbations or who fail to respond to standard asthma management. 

The goals of management of ABPA are to reduce inflammation, prevent exacerbations, and limit progression to end-stage lung disease. There have been a limited number of randomized controlled trials evaluating the treatment of acute ABPA, so the optimal treatment regimen is not clear [3,4,5,6]. Systemic corticosteroids and azole antifungals are the mainstays of treatment of ABPA, but they carry a significant burden of side effects, and relapse is not uncommon when treatment is reduced [7]. Long-term use of azoles is associated with hepatotoxicity, electrolyte abnormalities, peripheral neuropathies, and phototoxic reactions [8]. Long-term corticosteroid use is associated with many side effects, including weight gain, osteoporosis, aseptic joint necrosis, adrenal insufficiency, cataract formation, and hyperlipidemia [9]. The majority of patients respond to therapy with corticosteroids, but up to 50% relapse when tapering the dose [10], while up to 25% do not respond to treatment with antifungals [7].

As not all patients respond to standard therapy, there has been increasing use of monoclonal antibody (MAB) therapies to target the immune response seen in ABPA [11]. 

The pathogenesis of ABPA involves activation of the innate immune system, causing a predominantly Th2 CD4+ cell response which generates an intense inflammatory reaction characterized by mast cell degranulation and an influx of large numbers of eosinophils and neutrophils [12]. IgE, interleukin (IL)-4, IL-5, and IL-13 play an important role in this pathway; therefore, agents such as omalizumab (anti-IgE), dupilumab (anti-IL4Rα), mepolizumab (anti-IL5), and benralizumab (anti-IL5R) have been used off-label as treatment options. Data on the use of these agents in ABPA are promising but are mainly based on observational studies and are relatively scarce. Omalizumab and benralizumab are generally well tolerated, with local injection site reactions, headache, and fatigue being the most commonly reported adverse reactions; allergic or anaphylactic reactions have been reported, although these are rare [7]. Herein, we report three cases of ABPA treated with omalizumab followed by benralizumab. These cases highlight the challenges of treating ABPA with standard therapy and support a potential role for biologic agents in the management of ABPA.

## 2. Case Series

### 2.1. Case 1

A 50-year-old woman with previously well-controlled asthma was assessed in a severe asthma clinic due to increased respiratory symptoms. She reported increased dyspnea, cough, and wheeze and had experienced six exacerbations requiring systemic corticosteroid therapy over the preceding six-month period. She was a current smoker with a background history of rhinosinusitis, osteoporosis, cataracts, and non-alcoholic steatohepatitis, and had no history of eczema or hay fever. She was prescribed high-dose fluticasone/salmeterol, tiotropium, theophylline, montelukast, and salbutamol for asthma control. On examination, auscultation of her chest revealed bilateral inspiratory and expiratory wheeze. Her asthma control test (ACT) score was 5. Spirometry revealed an obstructive defect with significant reversibility: Forced Expiratory Volume in 1 s (FEV1) 1.04 L (42% predicted), Forced Vital Capacity (FVC) 2.46 L (82% predicted), FEV1 increased 290 mL (27%) post bronchodilator. Chest CT demonstrated mild airway wall thickening with no evidence of bronchiectasis or emphysema. Laboratory investigations showed an eosinophil count of 0.7 × 10^9^/L (normal range: 0–0.5 × 10^9^/L) but otherwise unremarkable blood count and metabolic profile. The total immunoglobulin E (IgE) level was 6263 IU/mL, and the level of specific immunoglobulin E (IgE) to *A. fumigatus* was 44.40 IU/mL (normal: <0.35 IU/mL). She was diagnosed with allergic bronchopulmonary aspergillosis-seropositive (ABPA-S) using the ISHAM diagnostic criteria [1].

Considering her pre-existing steroid complications, this patient was initially commenced on a medium-dose corticosteroid regimen of prednisolone starting at 0.5 mg/kg daily tapering over 6 months but had no significant improvement in symptoms or total IgE level. Four weeks post the commencement of therapy, her IgE level was 5470 IU/mL, her eosinophil count was 0.3 × 10^9^/L, and her ACT score was 11. Voriconazole (200 mg twice daily) was then added 1 month into her treatment regimen; however, she developed derangement in liver function tests with GGT increasing from 252 IU/L prior to treatment to 676 IU/L. In addition, she experienced little clinical improvement with the addition of voriconazole and continued to experience daily asthma symptoms of cough and nocturnal dyspnea (ACT 10), as well as exacerbations approximately every 2 months, so voriconazole was discontinued. Treatment was then escalated, and she was commenced on anti-IgE therapy with omalizumab, which was administered at a dose of 375 mg every 2 weeks in hospital. This resulted in clinical improvement, with a reduction in salbutamol reliever use (four times per day to once per day) and a reduction in exacerbation frequency with no exacerbations in the 6-month period after commencing omalizumab. Furthermore, her total IgE level reduced to 866 IU/mL (Figure 1), her eosinophil count was 0.3 × 10^9^/L, and maintenance corticosteroid therapy was weaned and discontinued. Six months later, she was switched to benralizumab (30 mg every 8 weeks) to decrease the frequency of her visits to hospital. Since commencing benralizumab, she has remained well with no requirement for rescue or maintenance systemic corticosteroids or hospital admission in the last 18 months. Her day-to-day symptoms have improved and her most recent ACT score was 14. She has not experienced any adverse events while on biologic therapy.

### 2.2. Case 2

A 68-year-old man was referred to our service for assessment of his asthma and ABPA. He was an ex-smoker with a 20-pack-year history, and his past medical history also included alpha-1 antitrypsin deficiency (AATD), chronic obstructive pulmonary disease (COPD), eczema, hypertension, and gout. He was prescribed high-dose inhaled fluticasone propionate, indacterol/glycopyrronium, salbutamol, and low-dose prednisolone (5 mg) for asthma control. In addition, he received monthly intravenous alpha 1 antitrypsin augmentation therapy for AATD and pulsed methylprednisolone given 3 days per month for ABPA at an outside hospital. 

Twelve months prior to referral, ABPA with central bronchiectasis (ABPA-CB) was diagnosed based on high total IgE (1030 IU/mL), positive specific IgE to *A. fumigatus* (3.71 IU/mL), positive *Aspergillus* IgG, and thorax CT showing central bronchiectasis. His FEV1 was 1.09 L (34% predicted) and his FVC was 2.38 L (58% predicted). At diagnosis, his ABPA-CB had been treated with a high-dose corticosteroid regimen of monthly intravenous methylprednisolone infusions for 6 months. After 6 months of steroid therapy, he was also commenced on a 6-week course of itraconazole to reduce his steroid requirement. His total IgE reduced to 590 IU/mL and his steroid dose was tapered to a maintenance dose of 5 mg prednisolone. He continued to have ongoing symptoms of poorly controlled asthma, and in the year prior to his assessment he had five exacerbations requiring prolonged hospital admissions or escalation of corticosteroid treatment. On assessment, he was found to have chronic active ABPA based on clinical symptoms, raised total IgE (1639 IU/mL), and raised eosinophils at 0.8 × 10^9^/L. He was initially treated with 0.5 mg/kg prednisolone, reducing to his baseline dose of 5 mg over 3 months. He initially improved symptomatically and his total IgE reduced to 856 IU/mL with an eosinophil count of 0.0 × 10^9^/L. However, as the steroid dose was reduced, he reported increasing dyspnea, reduced exercise tolerance, and IgE increased to 1338 IU/mL (Figure 2), while his eosinophil count was 0.2 × 10^9^/L. He was re-treated with a 3-month course of voriconazole and then started on omalizumab, which was administered at a dose of 375 mg every 2 weeks in hospital. He remained on omalizumab for 5 months; his symptoms improved and he had only one exacerbation in that time. He was then switched to the anti-IL5R monoclonal antibody benralizumab (30 mg every 8 weeks) to decrease the frequency of his visits to hospital. Since commencing benralizumab, he has remained well with no requirement for systemic steroids or hospital admission in 12 months on treatment. He reports that his daily symptoms are much improved and he rarely requires his reliever inhaler. He has not experienced any adverse events while on biologic therapy.

### 2.3. Case 3

A 59-year-old man was referred for assessment at our severe asthma clinic with a history of asthma since childhood, obstructive sleep apnea, and ABPA-CB. He was prescribed montelukast, budesonide/formoterol, tiotropium, and salbutamol for asthma control. He complained of dyspnea on exertion and chronic sputum production. He had been diagnosed with ABPA 7 years previously. This was treated initially with a course of oral corticosteroids and remained quiescent for 4 years, at which time he had a hospital admission with a severe exacerbation of ABPA. He was acutely unwell with an oxygen saturation of 86% on room air, respiratory rate of 35 breaths per minute, and heart rate of 120 beats per minute. At this time, his total IgE was 8939 IU/mL, eosinophils were 1.8 × 109/L, and *Aspergillus* precipitins were positive. His FEV1 was 1.16 L (45% predicted), his FVC was 3.54 L (117% predicted), and a chest X-ray showed extensive background parenchymal changes with dense patchy consolidations in the right lower and mid zones. He complained of dyspnea, wheeze, and sputum production that had not responded to escalation of his asthma treatment. A diagnosis of relapsed ABPA was made and treated with a medium-dose tapering corticosteroid regimen of prednisolone starting at 0.5 mg/kg and a 6-month course of voriconazole. His total IgE reduced to 3195 IU/mL 2 months later and his eosinophil count was 0.0 × 10^9^/L. After voriconazole was discontinued, he developed increased dyspnea and sputum production within 1 month. Voriconazole and oral corticosteroid were restarted and continued for another 6 months. After treatment with voriconazole, his total IgE was 986 IU/mL (Figure 3) and his eosinophil level was 0.3 × 10^9^/L on a maintenance dose of 5 mg prednisolone. Unfortunately, when voriconazole was discontinued a second time, his symptoms of exertional dyspnea and wheeze returned, and he was referred to our clinic for further assessment. On assessment, he had ongoing dyspnea and sputum production and had experienced five exacerbations requiring systemic corticosteroids in the previous 12 months. His total IgE was 5406 IU/mL and his eosinophil level was 1.0 × 10^9^/L. Specific IgE to *A. fumigatus* was positive and *Aspergillus fumigatus* precipitins were positive. His ACT score was 8. His treatment was escalated to omalizumab. He responded well and reported an improvement in his symptoms, his exercise tolerance was much improved, and his ACT score increased to 15. He continued to experience mild dyspnea on exertion and cough and had two exacerbations requiring oral corticosteroid treatment in the 5 months following commencement of omalizumab. Considering the ongoing symptoms and exacerbations, the patient was switched to benralizumab (30 mg every 8 weeks). At an assessment 2 years after starting benralizumab, his ACT score was 18 and his exacerbation rate had improved to one per year. He reports an improvement in daily symptoms, improved sleep, and a reduction in his reliever inhaler use. He has not experienced any adverse events while on biologic therapy.

## 3. Discussion

ABPA develops in susceptible hosts who repeatedly inhale *Aspergillus* spores; defective clearance of conidia in the airways allows them to germinate into hyphae and secrete proteolytic enzymes. These can activate and damage the airway epithelium and lead to activation of the innate immune system, causing a predominantly Th2 CD4+ cell response [13]. This involves the release of interleukin (IL)-4, IL-5, IL-13, CCL17, IL-9, and others [12]. The Th2 response generates an intense inflammatory reaction characterized by mast cell degranulation and an influx of large numbers of eosinophils and neutrophils [14]. Persistent inflammation leads to bronchiectasis, and if the disease remains uncontrolled, pulmonary fibrosis and end-stage respiratory disease can occur.

The goals of treatment are controlling inflammation, reducing the number of exacerbations, and limiting the progression of lung damage. Oral corticosteroids are the mainstay of therapy; however, they carry a significant side-effect burden when used in the long term. Additionally, up to 50% of patients relapse when the steroid dose is tapered, and 20–40% of patients remain steroid-dependent [10]. Azole antifungal agents are also used in patients who require high doses of steroids or have difficulty reducing their steroid dose, with the aim of reducing the level of antigen and thereby reducing the inflammatory response. Itraconazole and voriconazole have been used, with voriconazole having better gastrointestinal tolerability and bioavailability.

Randomized controlled trials of itraconazole showed a response rate ranging from 46% [15] to 88% [4]. However, adverse events are seen frequently—39% in one study [16]. A case series from 2012 using voriconazole and posaconazole demonstrated clinical response in 75% of patients on voriconazole and 78% of patients on posaconazole [17]. In that same study, 20% of patients on voriconazole developed significant adverse events requiring a switch to posaconazole [17]. Of the seven patients who discontinued voriconazole, four experienced a relapse within 12 months [17]. Antifungal therapy is a useful adjunct in the treatment of ABPA, but there is certainly a significant number of patients who do not respond to azole therapy, and relapse after discontinuation of therapy is common.

As many patients do not respond to standard care, the use of biological drugs for the treatment of ABPA has been increasing in recent years, though this is an off-label use. Biologic agents inhibit some of the fundamental pathways for the development of ABPA. Omalizumab was the first biologic agent used off-label in ABPA as it binds to free circulating IgE and inhibits the binding of IgE to receptors on mast cells and basophils, blocking the IgE-mediated secretion of inflammatory mediators from these cells. While this was an attractive approach, the levels of free circulating IgE often far exceed the binding capacity of omalizumab at its highest licensed dose, resulting in unbound IgE continuing to circulate and interact with its effector cells. Furthermore, the strategy of treating ABPA with an anti-IgE MAB does not address the unopposed effects of IL-5 and IL-13 produced by Th2 and ILC-2 cells. High levels of IL-5 result in increased numbers of eosinophils recruited to the airways and prolonged survival of these eosinophils at the site of inflammation. High levels of IL-13 promote mucus hypersecretion through both goblet cell hyperplasia and mucus gland hypertrophy, which may promote the survival of *Aspergillus* within the airway. This provides a biologic rationale to treating these patients with biologic therapies that target the IL-5 and IL-13 pathways.

To date, there has been a lack of randomized controlled trials to support the use of biologic agents, but there have been several case reports and case series. A literature review in 2020 included 32 studies and 161 patients in total; 60% of the studies analyzed omalizumab use, and the remaining studies were distributed among the rest of the biologics [11]. With regard to omalizumab, 40% of patients had a significant reduction in IgE post treatment (>35%), 66% had a reduction in their steroid dose, and 95% had a reduction in exacerbation frequency. Two case reports of patients treated with benralizumab were included. Both cases showed a clinical improvement; however, the follow-up time was short, and not all variables were described. Nine studies including 32 patients who were treated with mepolizumab were reviewed; in these studies, 90% of patients were able to discontinue steroids, and the remaining 10% had a dose reduction to a dose between 2.5 mg and 5 mg of prednisolone. A reduction in IgE of 66.5% from baseline was shown in the four patients for whom pre- and post-treatment IgE was reported. In patients treated with dupilumab, all patients had a reduction in total IgE levels, and 20 out of 21 patients reported an improvement in exacerbation frequency. This review highlighted that although most patients responded with a decrease in total IgE levels, there were cases where the IgE did not reach the 35% decrease; however, there was significant clinical improvement, suggesting that patient-centered outcomes are crucial for monitoring response.

The patients described in our case series presented in different stages of ABPA; the patient in Case 1 had a relatively early presentation before the development of central bronchiectasis, Case 2 describes a case of chronic ABPA with persistent symptoms, and Case 3 describes an acute life-threatening relapse of ABPA on a background of remitting disease. Despite variances in their presentation, these cases were all treated with a similar regimen and had similar responses to therapy. These patients had all received standard therapy with systemic corticosteroids and at least one course of voriconazole; however, each had persistence or relapse of ABPA despite this. These patients had a significant daily symptom burden, as well as frequent exacerbations. Treatment with omalizumab alleviated symptoms and significantly reduced their exacerbation rate. Given the extremely high IgE levels seen in ABPA, the maximum dose of omalizumab was indicated in these patients, requiring hospital visits every 2 weeks for administration. This frequency of hospital visits represents a significant burden both to the patients and to the healthcare system. In the setting of the COVID-19 pandemic, and taking into account the clinical context of these three patients, the decision was made to switch these patients to benralizumab to allow for home administration and reduce hospital visits. Each of these patients remained well after switching to benralizumab, with no increase in exacerbations or symptoms. Dupilumab was not available for administration in Ireland during the management of these patients.

## 4. Conclusions

ABPA is an important consideration in patients with asthma who experience frequent exacerbation despite standard management. A high level of suspicion is required to enable early diagnosis and prevent disease progression and lung damage.

Biologic agents are useful in patients who are refractory to treatment with steroids and antifungal agents, those who experience side effects to steroids and antifungals, or those in whom these drugs are contraindicated. These case reports support the use of omalizumab and benralizumab in the treatment of ABPA, but further studies are required to evaluate the effectiveness of these medications. Longer follow-up periods and objective measurements of the impact of treatment are necessary.

## Figures and Tables

**Figure 1 medicina-58-00053-f001:**
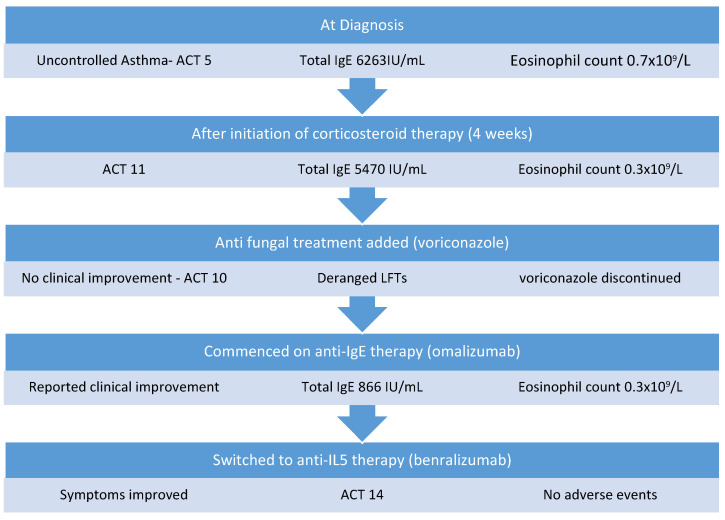
Management of Case 1.

**Figure 2 medicina-58-00053-f002:**
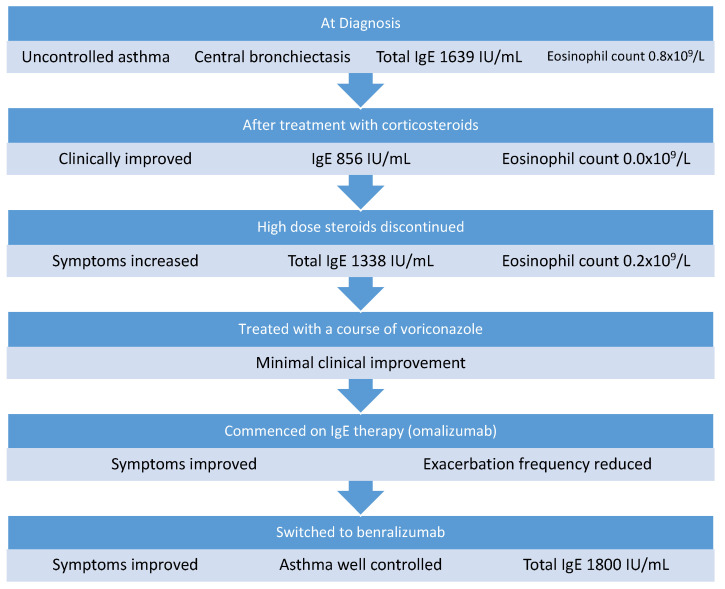
Management of Case 2.

**Figure 3 medicina-58-00053-f003:**
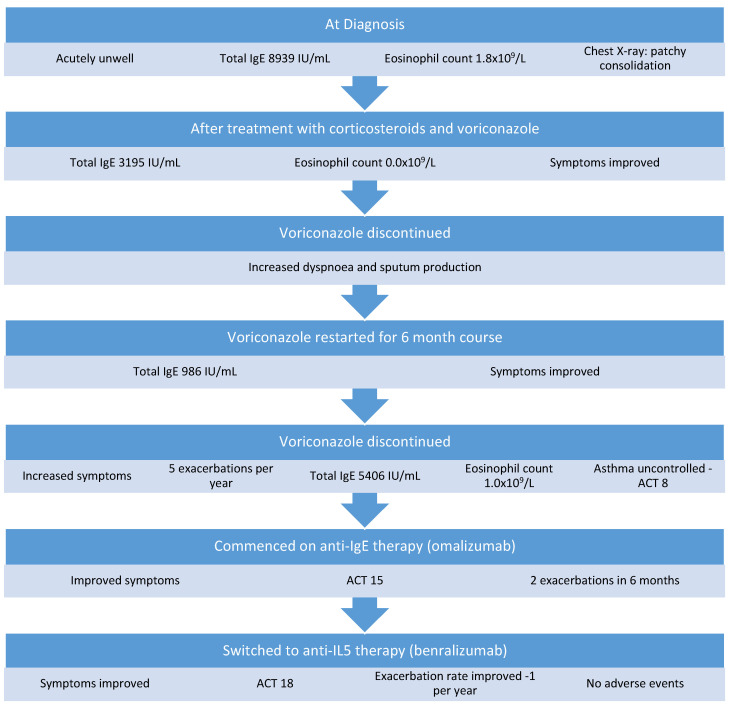
Management of Case 3.

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
