# Peer review of "The Use of Targeted Monoclonal Antibodies in the Treatment of ABPA—A Case Series"

_medicina, 2021, doi:10.3390/medicina58010053_

Round 1

Reviewer 1 Report

I have read the article by O’Reilly and Dunican with great interest. The authors presented 3 patients with ABPA who were treated with biologics.

Comments:

  • Case 2. Was he smoker? Was COPD diagnosis correct in his case?
  • Cases 2 and 3. Did they experience any side effect of biologics?
  • 2nd paragraph. Please list the most common side effects of antifungals and systemic steroids.
  • Please add the potential side effects of omalizumab and benralizumab.
  • It would be important to add blood eosinophil counts in parallel with IgE so that the readers could have an understanding if eosinophilia and high IgE are interrelated or if they represent different modalities of airway inflammation in ABPA.

Author Response

Thank you for your comments:

  • Case 2. Was he smoker? Was COPD diagnosis correct in his case?
  • Yes, I have added his smoking history, he was felt to have both asthma and COPD
  • Cases 2 and 3. Did they experience any side effect of biologics?
  • I have added in that they had no side effects
  • 2nd paragraph. Please list the most common side effects of antifungals and systemic steroids.
  • I have added this
  • Please add the potential side effects of omalizumab and benralizumab.
  • I have added this
  • It would be important to add blood eosinophil counts in parallel with IgE so that the readers could have an understanding if eosinophilia and high IgE are interrelated or if they represent different modalities of airway inflammation in ABPA.
  • I have added this

Reviewer 2 Report

Manuscript ID: medicina - 1503508

Type of manuscript: Case series

Title: The use of targeted monoclonal antibodies in the treatment of ABPA – a case series.

The article is focused on 3 cases of patients suffering from allergic bronchopulmonary aspergillosis that complicated course of asthma and COPD. I found this article interesting. Topic is very important, especially for clinicians.

I suggest only minor correction to the paper that may improve its quality and ease of reading.

Minor:

  • Please check spaces and dots through the text. Missing spaces before abbreviations or references in parentheses, lack of dot at the end of sentence etc. is very common.
  • I miss the clearly stated purpose of this publication. A sentence: “We report 3 cases of ABPA treated with omalizumab followed by benralizumab” doesn't get it done.
  • Page 4. Paragraph 2.3 Case 3. Second line; Did you really mean “obstructive sleep asthma”?
  • I also suggests a table, or figure, showing the treatment process, its effects and the characteristics of each case.

Author Response

Thank you for your comments:

1. I have checked spaces and dots and amended

2. I have added a statement to the introduction with regard to purpose

3. Page 4, paragraph 2.3: thank you , I have changed to obstructive sleep apnoea

4. I have added a figure showing treatment and outcomes